# Numerical Assessment of Terrain Relief Influence on Consequences for Humans Exposed to Gas Explosion Overpressure

**Yurii Skob** [1,*], **Sergiy Yakovlev** [2] , **Kyryl Korobchynskyi** [1] and **Mykola Kalinichenko** [3]

1    Mathematical Modelling and Artificial Intelligence Department, National Aerospace University "Kharkiv Aviation Institute", 61070 Kharkiv, Ukraine
2    Institute of Information Technology, Lodz University of Technology, 90-924 Lodz, Poland
3    Aircraft Engine Manufacturing Department, National Aerospace University "Kharkiv Aviation Institute", 61070 Kharkiv, Ukraine
*    Correspondence: y.skob@khai.edu

**Abstract:** This study aims to reconstruct hazardous zones after the hydrogen explosion at a fueling station and to assess an influence of terrain landscape on harmful consequences for personnel with the use of numerical methods. These consequences are measured by fields of conditional probability of lethal and ear-drum injuries for people exposed to explosion waves. An "Explosion Safety®" numerical tool is applied for non-stationary and three-dimensional reconstructions of the hazardous zone around the epicenter of the explosion of a premixed stoichiometric hemispheric hydrogen cloud. In order to define values of the explosion wave's damaging factors (maximum overpressure and impulse of pressure phase), a three-dimensional mathematical model of chemically active gas mixture dynamics is used. This allows for controlling the current pressure in every local point of actual space, taking into account the complex terrain. This information is used locally in every computational cell to evaluate the conditional probability of such consequences for human beings, such as ear-drum rupture and lethal outcome, on the basis of probit analysis. To evaluate the influence of the landscape profile on the non-stationary three-dimensional overpressure distribution above the Earth's surface near the epicenter of an accidental hydrogen explosion, a series of computational experiments with different variants of the terrain is carried out. Each variant differs in the level of mutual arrangement of the explosion epicenter and the places of possible location of personnel. The obtained results indicate that any change in working-place level of terrain related to the explosion's epicenter can better protect personnel from the explosion wave than evenly leveled terrain, and deepening of the explosion epicenter level related to working place level leads to better personnel protection than vice versa. Moreover, the presented coupled computational fluid dynamics and probit analysis model can be recommended to risk-managing experts as a cost-effective and time-saving instrument to assess the efficiency of protection structures during safety procedures.

**Keywords:** gas mixtures; explosion pressure wave; overpressure; impulse; probit analysis; probit function; negative impact conditional probability



## 1. Introduction

It is well known that hydrogen is one of the most explosive gases [1]. Therefore, increasing the use of hydrogen in the industry creates high risks of accidents, which lead to severe social and economic consequences [2]. Even nonsignificant violations of safety precautions or accidental equipment failures can cause hydrogen release into the atmosphere, mixing with air, causing the formation of a flammable gas mixture and an explosion that generates pressure waves propagated away from the accident epicenter (Figure 1) [3,4]. Sometimes, a wave propagation regime may change to detonation [5].

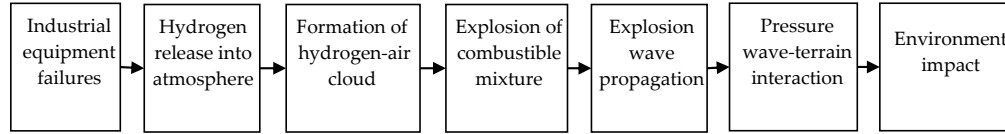

**Figure 1.** The development of a technogenic accident.

One of the most important and difficult-to-model phases of this complex physical process is the release of a dangerous admixture into the atmosphere. A comprehensive review of mathematical models of gas release during different industrial processes is represented in [3]. The hazardous zones are reconstructed from the fields of explosive concentrations for hydrogen and propane. The high-resolution computational fluid dynamic (CFD) models for flammable gas emissions provide noninvasive and direct quantitative evidence that may influence the safety procedures prepared by regulatory agencies in refining the safety limits in a cost-effective and time-saving manner [3]. A flammable gas mixture is considered as a potential hazard that can explode, with a potential shock-impulse impact to the environment. In our study, the release process is actually omitted, and a premixed hemispheric stoichiometric hydrogen cloud is considered in order to concentrate only on explosion pressure wave generation, its propagation through space with different reliefs, and probable interaction with humans at specific locations that can cause severe injuries for them.

Explosion waves make a shock-impulse impact on the environment, threatening the life and health of industrial workers, destroying infrastructure, and damaging equipment placed at industrial sites. Because of such accidents, social, material, and financial losses can be of catastrophic proportions.

In order to ensure the safety of working conditions on industrial sites, it is necessary to develop and apply protective equipment that can prevent or reduce, to an acceptable level, the possible harmful consequences caused by hydrogen–air explosions [4]. The effectiveness of these protection methods can be tested experimentally [6]. However, a full-scale physical experiment with a hydrogen explosion is difficult to implement, cumbersome, and too expensive. That is why a computational experiment based on computer information systems [7] implementing the considered accident scenarios (Figure 2) is widely used in practice. Thus, an engineering problem of mathematical modeling of physical processes of the considered emergency scenario is relevant.

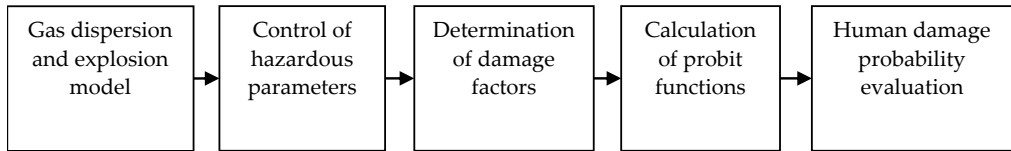

**Figure 2.** Accident consequences probabilistic evaluation scheme.

The scope of this research covers the problems of development of an environmental science technology based on computational fluid dynamics and probit analysis theory that can be used by safety experts to assess risk distribution around industrial objects where potentially dangerous flammable gases are used in technological processes. The *main aim* of this study is to numerically evaluate an influence of terrain landscape on the distribution of probable harmful consequences for personnel of a hydrogen fueling station caused by accidentally released and exploded hydrogen.

## 2. Review of the Literature

A mathematical model of the explosion of a hydrogen–air mixture cloud at a hydrogen fueling station site is considered in this paper. An influence of the terrain shape near the accidental hydrogen explosion in open space on the formation of a shock-impulse load and the resulting fields of the conditional probability of damage to working personnel

are analyzed. A state of the gas-dynamic environment at the site before an accident can be described as a set of normal values of overpressure, temperature, velocity vector, the chemical composition of the atmosphere. During an accidental explosion, these parameters become locally temporarily disturbed, and excess values of hazardous parameters form damaging factors that have harmful effects on the human body. Sometime after the accident, the environment returns to an unperturbed steady state again.

The purpose of this work is to use an effective mathematical model of the considered hydrogen explosion processes, for three-dimensional prediction and analysis of non-stationary distribution of damaging factors in order to determine the fields of the conditional probability of human damage based on probit analysis methodology.

An adequate description of the physical processes of dispersion of chemically reacting gases, mixing them with air, and further spreading the mixture into an open space [7], tunnel [8], or closed ventilated space [9] is possible only using the Navier–Stokes system of non-stationary equations for compressible gas [10]. Currently, numerical simulation of turbulent flows is carried out by solving the Reynolds–Favre-averaged Navier–Stokes equations, supplemented by a model of turbulence [11]. However, most turbulence models do not describe with an equal degree of adequacy the various types of flows that can appear [12]. This is especially true for currents with intense flow breaks and/or large pressure and temperature gradients.

In work [13], it is indicated that modern engineering methods for predicting the consequences of accidents on chemically hazardous objects (such as [14,15]) implement the Gauss model or the analytical solution of the mass transfer equation and do not take into account the blockage of the calculated space by impenetrable objects. The use of numerical kinematic models [16] to assess territorial risk is also limited to cases of impurity dispersion over a flat surface. Some papers take into account the complex terrain in the process of solving the mass transfer equation by the finite-difference method [13,15], but either there is no consideration for the three-dimensional nature of the flow around obstacles [13] or the effect of compressibility of the flow is not taken into account, which does not allow for using these mathematical models to calculate effects of all damaging factors (explosion shock wave load, thermal radiation, toxic dose), which may be present simultaneously during accidents.

In addition, modern techniques for assessing the technogenic impact on the environment are mainly based on a deterministic approach [17], and during probabilistic consequences assessment based on probit analysis, the table-view dependence of probability on the probit function is used for expert analysis [18]. It is not possible to apply this approach automatically in a computer system to obtain non-stationary fields of damaging factors and probability of damage, and it requires an improvement in computational methods and techniques.

Therefore, there is a need to build effective mathematical models and computational schemes for numerical modeling of three-dimensional flows of multicomponent gas mixtures, taking into account the complicated terrain shape in actual calculation space, compressibility, and chemical interaction effects, which allow for determining the full set of flow hazardous parameters for various scenarios of man-made accidents, calculating the damaging factors (including the shock-impulse load) and building space–time fields of human damage conditional probability needed to assess individual risk.

## 3. Problem Statement

Summarizing the literature review, we propose to solve the problem of assessment of how the terrain configuration around the epicenter of a gas explosion influences the safety situation at the technogenic object using a solution of the joint problem of gas dynamics of a chemically reacting gas mixture and the safety of a person who is under the influence of an explosion shock wave. For this purpose, firstly, the direct problem of flammable gas release and explosion is considered in an open space under normal environmental conditions using a three-dimensional system of equations that describes the motion of

the multicomponent chemically interactive gas mixture in the near-Earth atmosphere layer. It allows for obtaining time-dependent spatial information about the harmful-to-the-environment factor and shock-impulse distribution described by overpressure and impulse at the front of the explosion wave. Secondly, using the means of probit analysis, we can determine the conditional probability of the negative impact that causes the explosion to the environment in every control point of space at any moment of time.

Repeating this calculation process for different configurations of terrain relief, which kind of terrain is more harmful or safe for the personnel can be easily found, comparing the value of impact consequences in a specific working place. This information can be used during the process of determining where the best location of the technogenic object would be considered.

## 4. Materials and Methods

### 4.1. Method of Assessing the Impact Caused by an Explosion Wave

It is necessary to determine the peculiarities of the influence of terrain shape near an explosion accident on the spatial and temporal distribution of the shock-impulse load and the probability of personnel harm damage during an explosion of a hydrogen–air cloud at a fueling station site with a two-level landscape based on a mathematical model of the considered physical processes [19].

An accidental release of hydrogen at an industrial site is usually accompanied by the formation of a hydrogen–air mixture, which can explode under the influence of external factors. The resulting explosion wave spreads through the site, causing a shock-impulse load on humans and leading to harmful consequences for their health (Figure 1).

The harmful damaging impact of the shock wave according to a probabilistic assessment approach is determined by the maximum overpressure $\Delta P_+$ (relative to atmospheric pressure $P_0$) of the wave front and compression phase impulse $I_+$ (Figure 3).

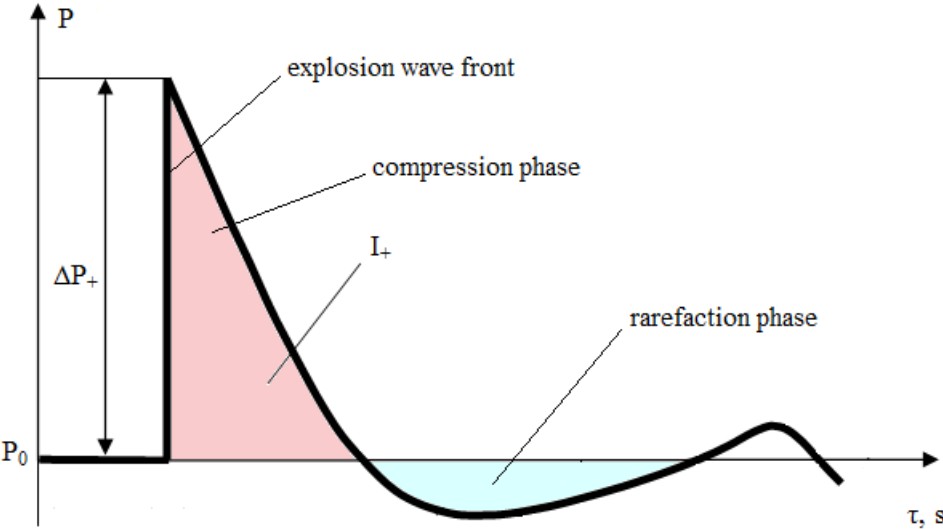

**Figure 3.** The typical profile of an explosion wave.

The values of these indicators in each control point can be used to determine the individual risk of negative impact on personnel. The risk assessment of the harmful effects of damaging factors on the human body at the accident site is one of the main stages of the safety analysis process of an industrial object. It allows for drawing conclusions about the acceptability of risk and for evaluating the effectiveness of protective facilities. The probability of a specific scenario for the development of an accident $P_s$ depends on the statistical probability of the occurrence of such an accident $P_a$ and the conditional injury probability of an affected person $P_c$, which can be obtained using mathematical modeling.

The conditional probability $P$ of harmful impact on a person that is under the influence of an explosion shock wave depends on the probit function $Pr$, which is the upper limit of a definite integral of the normal distribution law with mathematical expectation 5 and variance 1:

$$P = \frac{1}{\sqrt{2\pi}} \int_{-\infty}^{Pr} e^{-\frac{1}{2}(t-5)^2} dt, \tag{1}$$

where $t$ is an integral degree of impact.

For instance, the probability of human lethal damage caused by overpressure can be estimated by the following ratio [20]:

$$Pr_1 = 5 - 0.26 \ln\left[(17,500/\Delta P_+)^{8.4} + (290/I_+)^{9.3}\right] \tag{2}$$

The probit function for rupturing human eardrums depends on the level of overpressure only and can be found by the formula [21]:

$$Pr_2 = -15.6 + 1.93 \ln \Delta P_+ \tag{3}$$

In order to automate the computational process of analysis and prediction—the table of discrete values of the "probit function probability" that is usually used in engineering practice—this dependence is replaced by a generalized piecewise cubic Hermitian spline [22]. The characteristics of such a spline allow one to avoid possible oscillations of the approximated function in the intervals.

### 4.2. Explosion Mathematical Model and Calculation Algorithm

For a series of comparative computational experiments, in order to evaluate the influence of the two-level terrain shape on the distribution of the wave overpressure at the possible location of the working place, we use a mathematical model of an instantaneous explosion of hydrogen–air mixture [12–14].

It is assumed that the main factor influencing the physical processes under consideration is the convective transfer of mass, momentum, and energy. Therefore, it is sufficient to use the simplified Navier–Stokes equations, which are obtained by dropping the viscous terms in the mixture motion equations (Euler approach with source terms) [13].

The computational domain is a parallelepiped located in the right Cartesian coordinate system (Figure 4). It is divided into spatial cells whose dimensions are determined by the scale of the characteristic features of the area (roughness of streamlined surface, dimensions of objects).

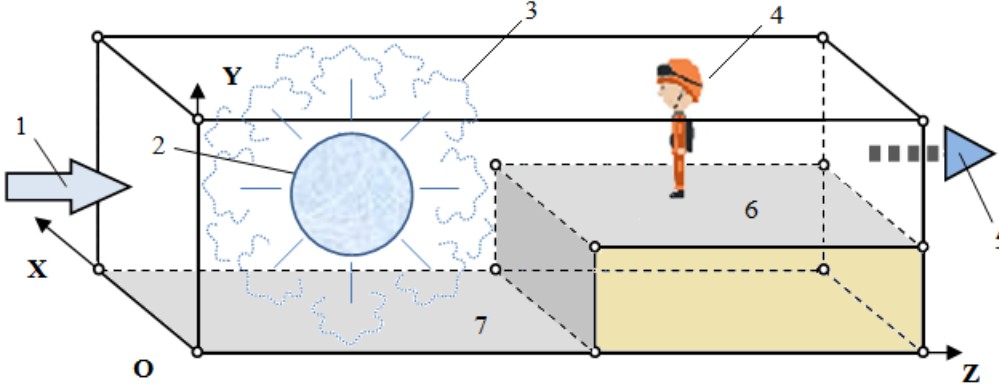

**Figure 4.** A computer model of the hydrogen–air cloud explosion: 1: inlet air; 2: hydrogen cloud; 3: combustion products; 4: personnel; 5: output mixture; 6: working place terrain level; 7: explosion terrain level.

According to the mathematical explosion model, the following boundary conditions are used: (1) at the entrance to the calculation area, total enthalpy, entropy function, and wind speed vector direction angles are set. Flow parameters here are determined with the involvement of the ratio for the "left" Riemann invariant; (2) at the exit from the calculation area, atmospheric pressure is set with the involvement of the ratio for the "right" Riemann invariant; (3) on the surfaces of solid bodies, "no flow" condition is set.

The following initial conditions are used: (1) in all gaseous "air" computational cells of the calculation area, the parameters of the atmosphere environment are set; (2) in all the cells occupied by the flammable cloud, the gas mixture flow parameters are set with relative mass concentration of the admixture $Q \leq 1$.

It is assumed that the instantaneous chemical reaction takes place in all elementary volumes of the computational grid, where the hydrogen concentration is within the limits of ignition ($Q_{min} \leq Q \leq Q_{max}$). This means that the parameters of the two-component mixture (air and fuel) in the control volume immediately obtain new values of the parameters of the three-component mixture (air, combustion products, and residues of fuel). In other words, it is assumed that the flame front propagates with infinite velocity [17].

A computer solution of the fundamental equations of gas dynamics for a mixture supplemented by the mass conservation laws of admixtures in the integral form is obtained using the explicit Godunov's method [23]. To approximate the Euler equations, the first-order finite-difference scheme is used. Central differences of second order are used for the diffusion source terms in the conservation equations of admixtures. Simple interpolation of the pressure is applied in the vertical direction. Godunov's method is characterized by a robust algorithm that is resistant to large disturbances of the flow parameters (e.g., pressure), which allows for obtaining a solution for modeling large-scale explosions of gas mixtures in calculation spaces of various types of configuration [24].

The mathematical model was validated with respect to Fraunhofer ICT experimental data for hydrogen and propane explosions [25].

The software "Explosion Safety®" (ES) [26] was used to analyze the explosion of a hydrogen cloud and dispersion of the combustion products processes, to forecast the pressure history at control points of human location and to evaluate safety differences between the various terrain options of the calculated space. The software can also be used to forecast the environmental impact of toxic spills [27]. It allows for calculating the density, velocity, pressure, temperature of the mixture, concentration of the mixture components (combustible gas, air, and combustion products), and the heat release rate within each control volume of the mixture at each discrete time step. The computer had the following characteristics: Intel® Core™ i7-360QM CPU @ 2.40 GHz 2.40 GHz, 16.0 Gb RAM, Windows 7. CPU time for each experiment was about 15 min.

## 5. Experiments

A computer simulation of the explosion of a cloud of the hydrogen–air mixture resulting from an accidental release from a destructed dispensing cylinder at a hydrogen fueling station is carried out. The calculated area is shown in Figure 5. The computational experiment is carried out at air velocity q = 0.0 m/s, ambient temperature 293 K, and pressure 101,325 Pa at the entrance to the considered area. The dimensions of the computational domain and other specific sizes are the following: the length $L_z$ = 31.2 m, the height $L_y$ = 14.0 m, the width $L_x$ = 20.2 m, and the height of the first ground level $Y_1$ = 4.0 m. The second-level part of the site begins from $Z_2$ = 13.2 m and has a changeable height H.

The cloud of the hydrogen is located at a distance of $Z_1$ = 10.1 m from the origin of the computational domain; the radius of the cloud is R = 2.88 m. Two control points P1 and P2 at the distances $Z_{p1}$ = 3.2 m and $Z_{p2}$ = 7.1 m from an explosion epicenter C are established. They are located in characteristic places of the second-level part of the industrial site, where overpressure history is monitored.

Five options of the design scheme V1–V5 are considered (Table 1) in order to assess the influence of the site terrain shape on overpressure and damage probability fields. The options differ only by the height H of the second terrain level.

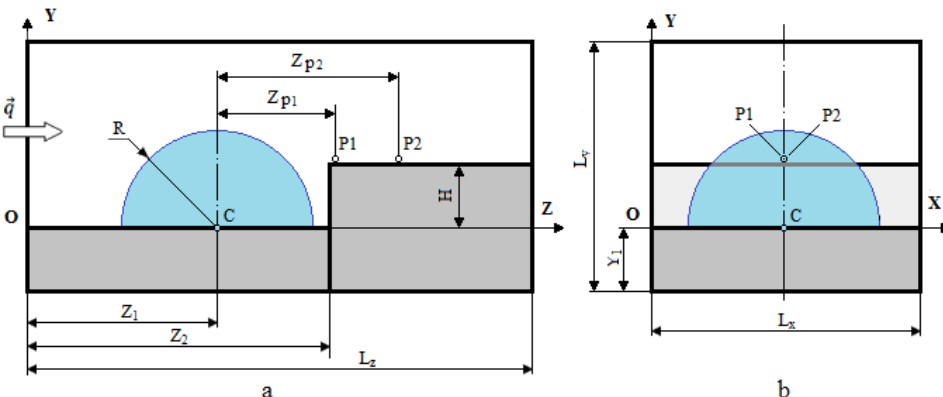

**Figure 5.** A scheme of the calculated area in planes YOZ (**a**) and YOX (**b**).

**Table 1.** Types of calculation scheme.

| Variants | V1 | V2 | V3 | V4 | V5 |
|---|---|---|---|---|---|
| Scheme | 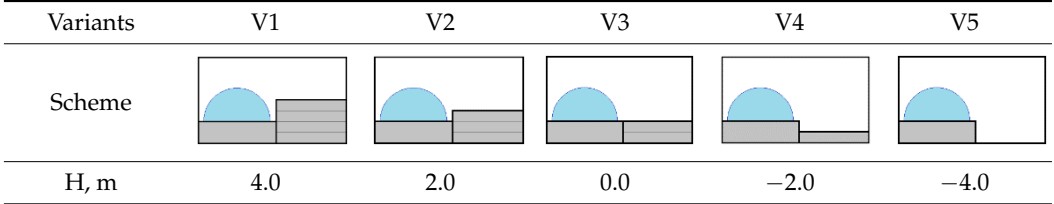 | | | | |
| H, m | 4.0 | 2.0 | 0.0 | −2.0 | −4.0 |

As a result of the hydrogen–air mixture explosion, a cloud of combustion products with high pressure and temperature is formed. The process of combustion products dispersion takes place. It is accompanied by shock wave propagation from an explosion epicenter. During the calculation process, it is possible to monitor the 3D pressure distribution (Figure 6) in order to collect all the needed information to calculate the damage probability fields.

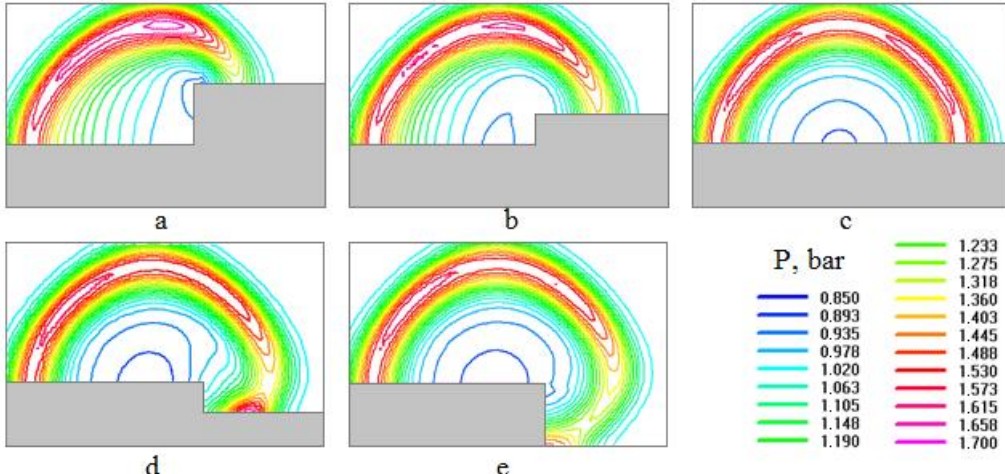

**Figure 6.** Pressure distribution in the plane YOZ at t = 0.01 s: (**a**–**e**) options V1–V6.

The overpressure history at the control points P1 and P2 for different design scheme options V1–V5 are presented in Figure 7. It is obvious that the most dangerous variant of the landscape terrain corresponds to variant V3, where both terrain parts of the industrial site are at the same level (height H = 0 m). Any other option of the calculation scheme leads

to a decrease in both maximum overpressure and compression phase area that means less shock-impulse loads on people standing in control points P1 and P2. This trend can be noticed also from the comparison of pressure distribution in plane XOY at some moment of time (0.01 s) after the explosion (Figure 6).

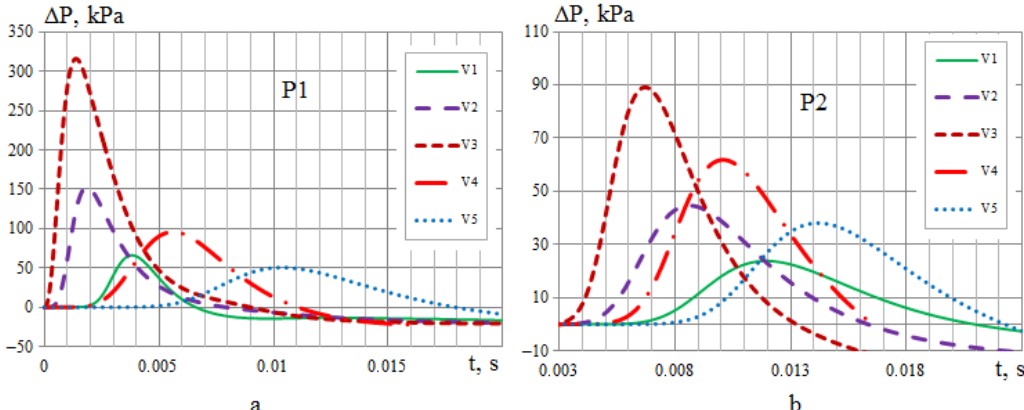

**Figure 7.** Overpressure history at the control points P1 (**a**) and P2 (**b**).

The collected data allow us to extract all the information needed to evaluate the damaging factors of the explosion shock wave (maximum overpressure (Figure 8) and compression phase impulse (Figure 9)) and to calculate the values of the conditional probability of lethal consequences (Figure 10) according to formula (2) as well as eardrum rupture (Figure 11) according to formula (3) at control points P1 and P2 for different terrain options V1–V5.

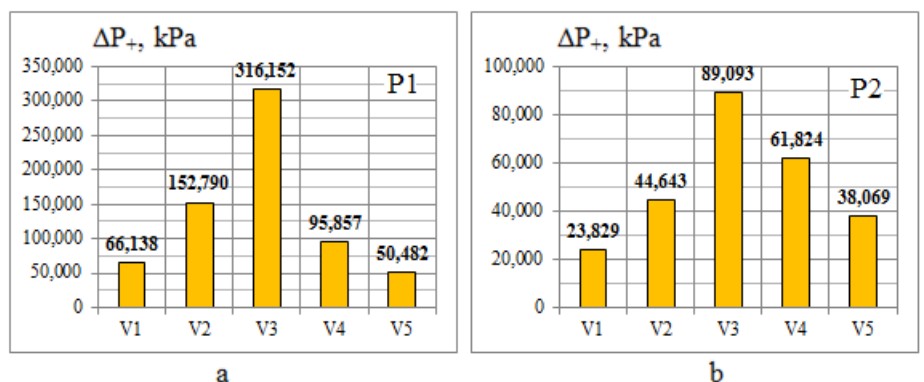

**Figure 8.** Maximum overpressure at control points: (**a**) point P1; (**b**) point P2.

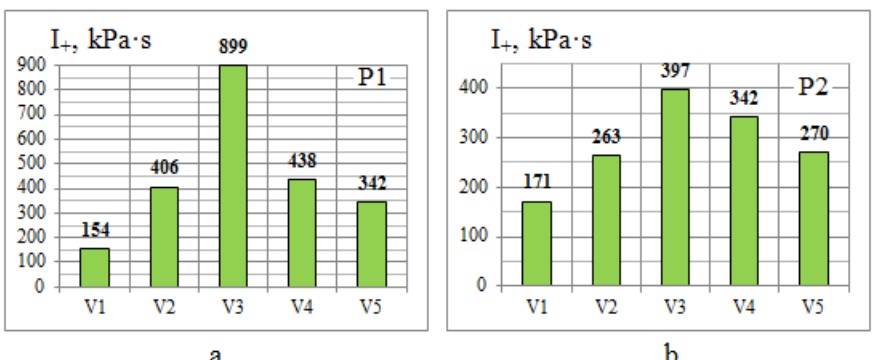

**Figure 9.** Compression phase impulse $I_+$ at control points: (**a**) point P1; (**b**) point P2.

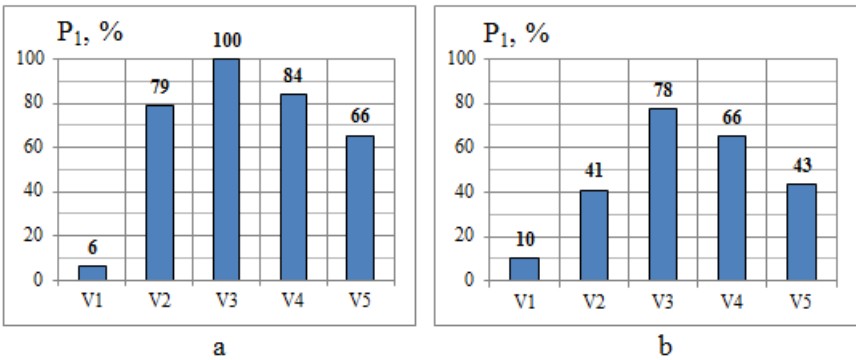

**Figure 10.** Lethal probability at control points: (**a**) point P1; (**b**) point P2.

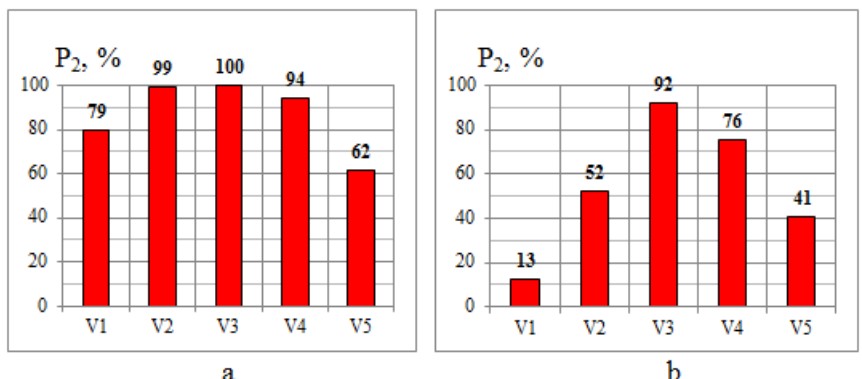

**Figure 11.** Eardrum rupture probability at control points: (**a**) point P1; (**b**) point P2.

The lethal consequence conditional probability in the most exposed to overpressure, vertically centered plane YOZ, and on the ground of the second part of the industrial site (possible working places location) in plane XOZ are displayed in Figures 12–16.

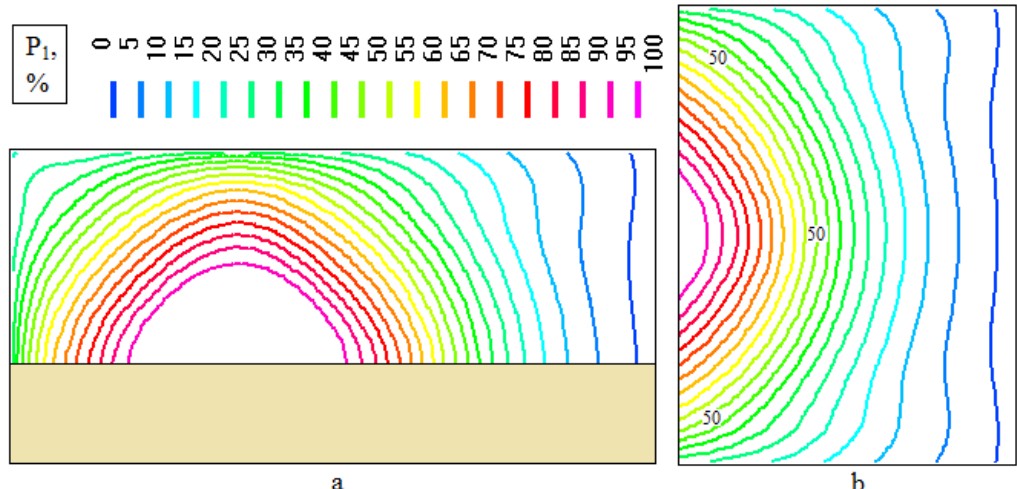

**Figure 12.** Lethal probability fields for option V3: (**a**) plane YOZ; (**b**) plane XOZ (a working place).

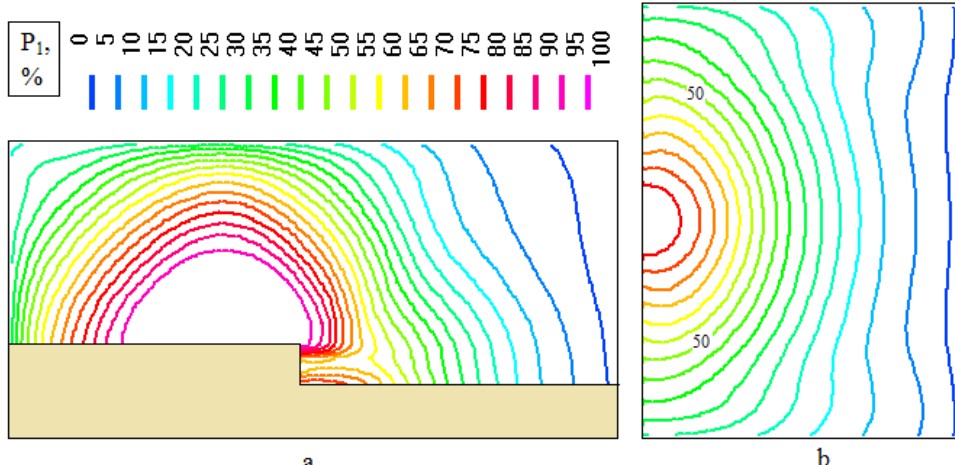

**Figure 13.** Lethal probability fields for option V4: (**a**) plane YOZ; (**b**) plane XOZ (a working place).

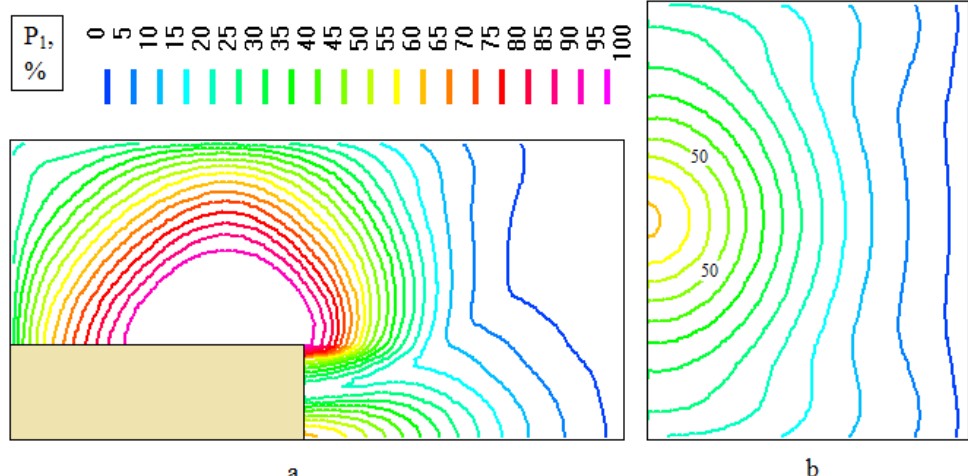

**Figure 14.** Lethal probability fields (option V5): (**a**) plane YOZ; (**b**) plane XOZ (level two).

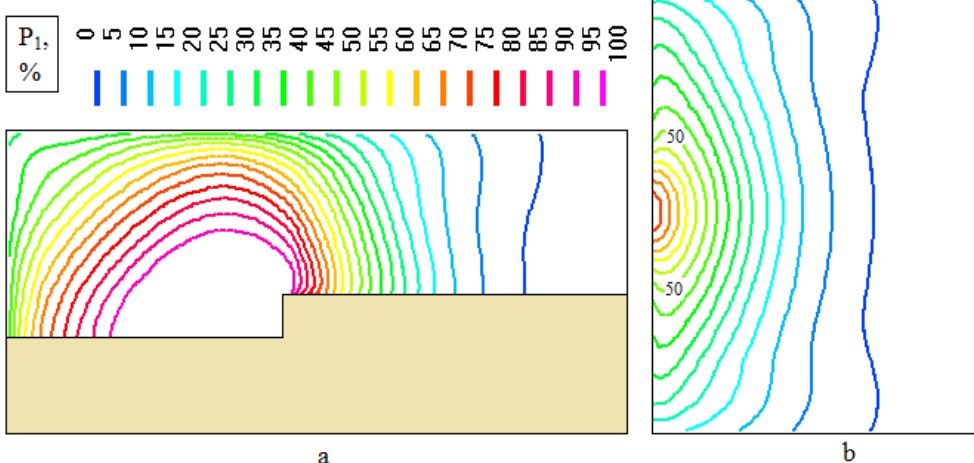

**Figure 15.** Lethal probability fields (option V2): (**a**) plane YOZ; (**b**) plane XOZ (level two).

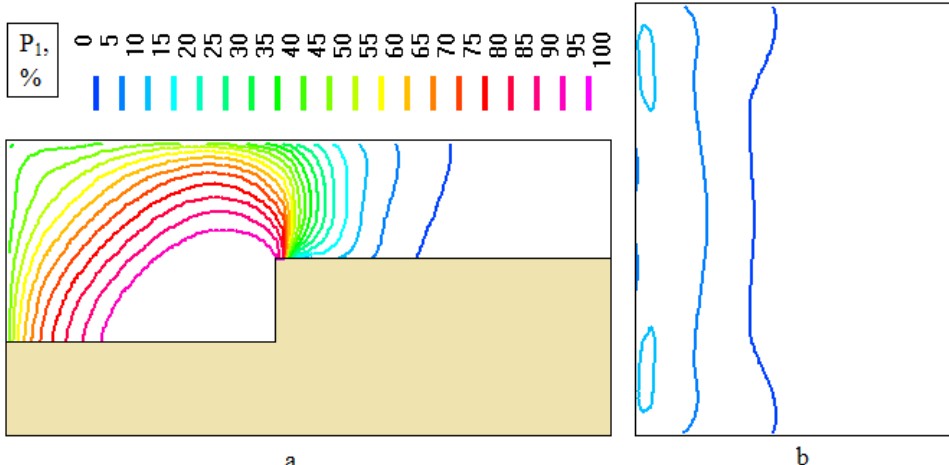

**Figure 16.** Lethal probability fields for option V1: (**a**) plane YOZ; (**b**) plane XOZ (a working place).

In order to compare different schemes of the terrain landscape, the area of zone $S_{50}$ on the surface of the second part of the industrial site where the lethal consequences conditional probability is greater than 50% (which is considered dangerous for humans) is calculated (Figure 17).

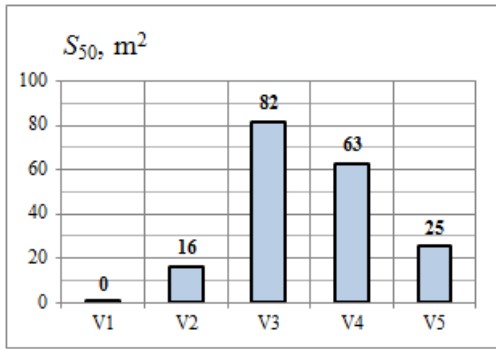

**Figure 17.** A dangerous area $S_{50}$ at the working place level.

## 6. Results

A numerical simulation of the pressure wave propagation from the epicenter of the stoichiometric hydrogen cloud explosion along space, with different Earth reliefs, enabled us to reconstruct the hazard zones for humans potentially located at two distant control points (Figure 5). Five options of an explosion epicenter and control point Earth levels were considered (Table 1): two options with deeper located explosion level, one evenly leveled option, and two options with deeper located control points. For all variants of relief, the following parameters were obtained:

- Pressure 3D field during explosion wave propagation (Figure 6);
- Overpressure history in control points P1 and P2 (Figure 7);
- Maximum overpressure of an explosion wave in control points (Figure 8);
- Impulse of explosion wave pressure phase in control points (Figure 9);
- Lethal (Figure 10) and ear-drum rupture (Figure 11) probability in control points;
- Lethal conditional probability 3D fields (Figures 12–16);
- Hazardous zone area at control points level where lethal probability is greater than 50% (Figure 17).

Non-stationary pressure distributions in vertical cross-section YOZ (Figure 6) can give us a clue as to why probable consequences for humans in control points behave in such a manner. Deepening the relief level of the accident epicenter related to control points level (Figure 6a,b) leads to higher obstacles in front of the pressure wave and a bigger reflecting

effect. Deepening the relief level of control points related to the accident epicenter level (Figure 6d,e) leads to break-away and reattachment to the Earth of the pressure wave and a bigger expansion effect. The most dangerous option is an evenly leveled relief (Figure 6c) because the control points are openly exposed to the shock-impulse load.

From previous pressure fields, overpressure history in control points P1 and P2 can be collected (Figure 7) and processed in order to calculate maximum overpressure of an explosion wave (Figure 8), which characterizes the shock effect for exposed humans and the impulse of the explosion wave pressure phase (Figure 9), which reflects the timespan of the load and assesses the probable consequences for humans as lethal (Figure 10) and ear-drum rupture (Figure 11) injuries. Lethal conditional probability 3D fields (Figures 12–16) help us to evaluate the dangerous zone area (starting from the edge of the control points level) (Figure 17), which can be used as an additional measure of relief variant safety characteristics.

It can be seen from the presented overpressure, impulse and consequences diagrams that the biggest shock loads correspond to option V3 with equal terrain levels of two parts of the industrial site. This leads to the highest values of the ear-drum rupture conditional probabilities (Figure 11) for both control points. Some decrease in shock loads in control point P2 (compared to point P1 at the edge of the level-two part) can be explained by the more distant location of this point from the explosion epicenter.

Any other design scheme of the level-two part gives a decrease in shock loads that is especially noticeable for control point P2 (Figures 7b, 8b and 11b). Options V4 and V5, which correspond to deeper levels of the part two terrain, give less protective effect for point P2 than for the corresponding options V2 and V1 with higher levels of the part two terrain. It can be explained that in options V2 and V1, part of the explosion wave meets an obstacle and is reflected backward. For control point P1 (Figures 7a, 8a and 11a), the deepening makes a bigger effect in options V4 and V5 comparable to options V2 and V1, maybe because of the less intensive expansion process around the convex corners of the terrain.

Very similar behavior can be seen in compression phase impulse distribution (Figures 7 and 9). Higher levels of the terrain (options V2 and V1) create bigger protection in control points from impulse loads than deeper levels (options V4 and V5).

The total effect from maximum overpressure and impulse loads can be clearly seen in Figure 10 for lethal probabilities in control points, and in Figures 12–16 for the fields of this consequence parameter. Higher levels of options V2 and V1 better protect humans than deeper variants of terrain, especially in point P2 (Figures 10b, 12b, 13b, 14b, 15b and 16b). This conclusion can be confirmed by such safety characteristics as an area $S_{50}$ of dangerously high values (> 50%) of the conditional lethal probability (Figure 17) on the surface of the level-two part of the industrial site (working place). It is clearly seen that higher-level variants V2 and V1 protect the working place much more effectively than deeper level variants V4 and V5 of landscape in relation to variant V3 with evenly leveled terrain.

## 7. Discussion

A large-scale field experiment [6] is the most adequate way to reconstruct hazardous zones of an accidental release and explosion of flammable gases at industrial objects of high risk. Experimentally measured flow parameters such as admixture concentration, temperature and pressure of explosion products can be used by safety experts to analyze and assess harmful consequences for humans and industrial constructions around an explosion epicenter. However, field experiments are very time-consuming and cost-ineffective, depend on environment conditions, and cannot really be used to carry out such series of experiments to compare different options of relief shape, as this study does. That is why mathematical modeling of release, dispersion, and explosion processes is an effective alternative way to obtain all the needed information about flow parameters with much wider opportunities to experiment with different environment conditions, landscape shape, flammable and/or toxic gases, and various scenarios of accidents. A validation of capabilities of the presented computational fluid dynamics model to reproduce a large-scale

hydrogen explosion in open atmosphere [28,29] against the results of intercomparison exercises on capabilities of other CFD models [7], evidence that the presented model adequately describes an explosion process and propagation of an explosion wave in open space. The CFD models analyzed in work [3] (LES, RANS, and FDS models) are based on Navier–Stokes equations. These models can be very useful while thoroughly predicting hydrogen distribution during release and dispersion processes (in open space [7], in tunnels [8], and with pressure relief vents [9]), but they consume huge computer resources and require careful selection of a turbulence model, which is different for different types of flow conditions. These advanced models give very similar results to our CFD model's results in predicting maximum overpressure and impulse at the shock wave front during a large-scale hydrogen explosion [28,29], which is crucially important in probit analysis to evaluate harmful consequences to the environment caused by the explosion. Our CFD model represents the Euler approach with source terms (simplified Navier–Stokes equations obtained by dropping the viscous terms in the mixture motion equations) [13], does not require turbulence model selection, consumes much less computer resources, and is very useful in comparing experimental series.

The presented methodology provides a mathematical tool to evaluate whether the differently leveled terrain at an industrial open space, where an accidental explosion of hydrogen takes place, can change the safety level for humans located near the epicenter of an explosion. The numerical analysis of a three-dimensional pressure field's history during an explosion's wave propagation enabled us to quantify the effect of different options of terrain shape on probable consequences for people exposed to explosion shock-impulse impact. With the use of probit analysis, incorporated into the CFD model, it was possible to present diagrams of conditional probability of harmful injuries to humans at work places during the explosion for different options of terrain. This was in line with work [11], where lethal probability was obtained on the base of results of fluent CFD modeling of hydrogen non-premixed combustion in an enclosure with one vent and sustained release. However, consequences for humans were assessed using "overpressure on impulse" diagrams that provide probability isolines, and this technique did not allow for building three-dimensional fields of impact probability and for making a transition to individual risk assessment in future safety evaluation processes.

*Limitations to the Study*

The presented methodology provides satisfactory results for open space explosion wave propagation but may encounter some problems in evaluating shock-impulse consequences in narrow tunnels and small premises where multiple explosion wave reflections take place, and it would be difficult to extract the impulse of the explosion wave from the model. The CFD model was used only for the assessment of harmful consequences zones induced by an explosion of a premixed hemispheric cloud, and only one flammable gas (hydrogen) was analyzed. In the future, we would like to consider other flammable gases and include into consideration an admixture releasing process before invoking an explosion. There could be different release scenarios involved, such as evaporation from a spilled liquid spot or jet emissions from a destroyed high-pressure storage vessel. The released admixture or explosion products were assumed to be nontoxic. In the future, we would like to consider a coupled scenario of dangerous zones formation (accidental explosion of flammable gas mixture and dispersion of toxic admixture) to predict the combined consequences for humans.

## 8. Conclusions

The purpose of this study was to designate risky zones for humans after an accidental explosion of a premixed stoichiometric hemispherical hydrogen cloud placed at differently leveled terrains. Currently, hydrogen is widely used in transport that requires refueling at filling stations where hydrogen is kept in high-pressure vessels. Therefore, it is important to be aware of risks of accidental hydrogen release, formation of a flammable mixture with

air, and its explosion with the generation of a shock wave that propagates along the ground surface, which can cause injuries to humans at working places. In order to determine these hazardous zones and probable environmental consequences in control points and to evaluate the influence of terrain shape on the scale of consequences, the ES software using computational fluid dynamics and probit analysis was applied. For the purposes of this study, a series of five simulations were made. They differed by two-level terrain options where the hydrogen cloud was placed at the first-level plane, and two control points (human locations) were placed at the second-level plane. The options of terrain configuration were compared on pressure three-dimensional field evolution during the explosion wave propagation, the values of maximum overpressure and impulse of the first pressure phase of the explosion wave, conditional probability for eardrum rupture and lethal outcome in control points, and the dangerous area value at the level-two plane where the lethal probability was greater than 50%. During every simulation, the same environment and hydrogen cloud parameters were applied.

It was obtained that higher-leveled working places in relation to the possible explosion epicenter terrain level could give better protection than deeper-leveled places.

In conclusion, incorporating the probit analysis procedure into the CFD model provides a powerful instrument to intercompare computer experiments, and it can be used by safety experts to develop measures to reduce the risk of considered accidents at industrial sites and to analyze the efficiency of protection structures. Further improvement of this methodology is possible in the direction of enhancing the accuracy of the gas-dynamics mathematical model and in considering a combination of accidental scenarios, taking into account various influencing factors.

**Author Contributions:** Conceptualization, Y.S. and S.Y.; methodology, Y.S. and S.Y.; software, K.K. and M.K.; validation, K.K. and M.K.; formal analysis, K.K. and M.K.; investigation, Y.S., K.K. and M.K.; resources, K.K. and M.K.; writing—original draft preparation, K.K. and M.K.; writing—review and editing, Y.S. and S.Y.; visualization, K.K. and M.K.; supervision, Y.S. and S.Y.; project administration, K.K. and M.K. All authors have read and agreed to the published version of the manuscript.

**Funding:** This research received no external funding.

**Institutional Review Board Statement:** Not applicable.

**Informed Consent Statement:** Not applicable.

**Data Availability Statement:** Generated data and test tasks are used.

**Conflicts of Interest:** The authors declare no conflict of interest.

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
