# Peer review of "Numerical Assessment of Terrain Relief Influence on Consequences for Humans Exposed to Gas Explosion Overpressure"

_computation, doi:10.3390/computation11020019_

Round 1
Reviewer 1 Report
1. The author could mention more about the contributions of practical applications in abstract and conclusion sections.
2. The authors could discuss more about the comparison with some traditional methods to highlight the advantages and contributions of the proposed method.
3. Please identify and mention the limitations of the current paper in conclusion section.
4. The manuscript should be proof read, otherwise it is difficult to understand and read. Please re-check the grammar and spelling.
Reviewer 2 Report
The paper called Numerical assessment of influence of terrain relief on environmental consequences caused by gas explosion by Yurii Skob, Sergiy Yakovlev, Kyryl Korobchynskyi, Mykola Kalinichenko. The paper is very good; there are only few small improvements to make. There are some major aspects I would like to highlight. There are some things that could be added to the paper to broaden the scope of the paper along with the group of potential readers.
Very good research work, requires a few additions and corrections;
1) The title should be redrafted to reflect the nature of the work.
2) In the abstract presented, the importance of the publication should be more concisely described including more extensive consideration of the methods, analyses and results of the research obtained.
3) It would be necessary to emphasize what is the main scope and purpose of the presented publication, it is difficult to find it in the text,
4) The chapter result and discussion should be separated because it is not known what is what,
5) In conclusion, it is worth specifying what is the direction of further research.
The presented conclusions may be of fundamental importance, therefore they should be presented in a better light and the author(s) should emphasize the original research contribution. I believe, that suggested amendments will significantly increase the relevance of the publication and will improve it. After applying all required changes, the paper is suitable for publication.
Reviewer 3 Report
The paper written by the following Authors: Yurii Skob, Sergiy Yakovlev, Kyryl Korobchynskyi, Mykola Kalinichenko, entitled “Numerical assessment of influence of terrain relief on environmental consequences caused by gas explosion” presents an interesting numerical study on the influence of terrain landscape on the distribution of probable harmful consequences to personnel of hydrogen fueling station caused by an accidentally released and exploded hydrogen.
Although the paper is interesting, I have some major concerns:
Title
The title reflects the results presented here.
Abstract
The abstract is lacking the aim of the study, material and methods description as well as an informative conclusion. It should be written in more details.
Introduction
In the introduction part Authors should add some overall information in paragraph/paragraphs dedicated on the hydrogen release with the use of numerical tools:
Processes 2021, 9(2), 307; https://doi.org/10.3390/pr9020307
Material and Methods
- There is no initial and boundary conditions for the mathematical model. It should be included in the manuscript.
- There is no information about the software applied for the calculations.
Results
This part should written in more details. Moreover, the is no discussion. Authors should refer to the literature from the same area.
Conclusions
Should be written in more details, referring to the results.
Round 2
Reviewer 2 Report
Thank you for the changes made.
Accept in present form.
Reviewer 3 Report
I accept the manuscript in present form.